# Distribution and Location Stability of the Australian Ophthalmology Workforce: 2014–2019

**DOI:** 10.3390/ijerph182312574

**Published:** 2021-11-29

**Authors:** Penny Allen, Belinda Jessup, Santosh Khanal, Victoria Baker-Smith, Kehinde Obamiro, Tony Barnett

**Affiliations:** 1Rural Clinical School, Hospitals’ Campus, University of Tasmania, Brickport Road, Burnie, TAS 7320, Australia; 2Centre for Rural Health, Newnham Campus, University of Tasmania, Launceston, TAS 7250, Australia; Belinda.Jessup@utas.edu.au (B.J.); kehinde.obamiro@utas.edu.au (K.O.); tony.barnett@utas.edu.au (T.B.); 3The Royal Australian and New Zealand College of Ophthalmology, 94-98 Chalmers Street, Surry Hills, NSW 2010, Australia; skhanal@ranzco.edu (S.K.); victoriabakersmith@ranzco.edu (V.B.-S.)

**Keywords:** ophthalmology, workforce distribution, rural

## Abstract

Objective: To investigate the ophthalmology workforce distribution and location stability using Modified Monash Model category of remoteness. Methods: Whole of ophthalmologist workforce analysis using Australian Health Practitioner Registration Agency (AHPRA) data. Modified Monash Model (MMM) category was mapped to postcode of primary work location over a six-year period (2014 to 2019). MMM stability was investigated using survival analysis and competing risks regression. Design: Retrospective cohort study. Setting: Australia. Participants: Ophthalmologists registered with AHPRA. Main outcome measures: Retention within MMM category of primary work location. Results: A total of 948 ophthalmologists were identified (767 males, 181 females). Survival estimates indicate 84% of ophthalmologists remained working in MMM1, while 79% of ophthalmologists working in MMM2–MMM7remained in these regions during the six-year period. Conclusion: The Australian ophthalmology workforce shows a high level of location stability and is concentrated in metropolitan areas of Australia. Investment in policy initiatives designed to train, recruit and retain ophthalmologists in regional, rural and remote areas is needed to improve workforce distribution outside of metropolitan areas.

## 1. Introduction

Socioeconomic deprivation, lower levels of educational attainment, lifestyle factors and chronic disease risk factors result in the higher prevalence of many eye diseases in rural and remote communities compared with metropolitan communities [1,2,3,4,5,6,7,8,9,10]. Australians living in rural and remote areas have been found to have higher prevalences of cataracts, pterygia and ocular trauma [1,2,3]. Indigenous Australians living in remote areas are more likely to be diagnosed with diabetic retinopathy [4,5], vision impairment and blindness [6,7]. Despite higher rates of eye disease and vision loss, people living in rural and remote Australia have poorer access to ophthalmologists [8] and cataract surgery [9,10].

Low levels of access to ophthalmology services in rural and remote communities are largely a result of the maldistribution of the workforce [11]. Innovative models, such as outreach services and shared care, have been implemented to address the inequities in access to ophthalmic care. Evaluations of these services have found the models reduce barriers to eye screening, treatments and surgery [12,13,14]; however, the overall size and distribution of the ophthalmology outreach workforce is yet to be mapped.

Whilst considerable research has been conducted into the distribution and retention of the general practitioner workforce, details about the workforce of specific specialties such as ophthalmology are scant. A 2018 Department of Health report noted that 84% of the ophthalmology workforce is located in metropolitan areas (Modified Monash Model category MMM1), although further detailed analysis of workforce distribution was not provided [11]. The Modified Monash Model (MMM) is how the Australian Department of Health defines whether a location is a city, rural, remote or very remote [15].

The shortage of doctors working in rural areas is an international issue. In Canada, a government report from 2012 found 20% of the population lived in rural areas, yet only 9.3% of doctors worked rurally [16]. In India, the problem of workforce maldistribution is particularly challenging with 66% of the population residing in rural areas yet only 33% of all health workers are located rurally [17]. The issue of workforce maldistribution is most acute amongst specialists, who tend to be more highly concentrated in urban areas than family physicians, as was reported by Barreto et al. in their study of rural counties in the United States [18]. The present study sought to address the paucity of information about the location stability of the ophthalmology workforce in Australia. Specifically, the aim was to investigate workforce distribution and location retention over time according to Modified Monash Model category.

## 2. Methods

The University of Tasmania Health and Medical Human Research Ethics Committee (Project ID: 23059) and RANZCO Human Research Ethics Committee provided approval for the research. De-identified data extracts listing all medical practitioners registered with the Australian Health Practitioner Registration Agency (AHPRA) were provided by AHPRA to the University of Tasmania Rural Clinical School. These data are stored in a centralised database (GradTRack, HREC reference: H0013913). The GradTrack steering committee provided permission for an extract of the AHPRA data containing ophthalmologists for the research.

Longitudinal locations were provided for a six-year period (2014 to 2019). The data set included all ophthalmologists registered with AHPRA for at least one year during this period. Data fields were: registration status, gender, primary training country, year of primary qualification, year of initial AHPRA registration and Primary Place of Practice (PPP) location (suburb and postcode) for each year. Registration status (current, failed to renew, surrendered, withdrawn, or deceased) was available for the years 2014–2018 but AHPRA did not provide this information for 2019. Ophthalmologists who did not have a current registration for at least one of the years were excluded.

MMM categories (based on the 2019 iteration) were mapped to postcodes of primary workplace location for each of the six years [15]. The MMM has seven categories of remoteness based on population size and remoteness from capital cities (MMM1—metropolitan, MMM2—regional centres, MMM3—large rural towns, MMM4—medium rural towns, MMM5—small rural towns, MMM6—remote communities and MMM7—very remote communities). Approximately 70% of the Australian population live in MMM1 areas [19]. In this paper, the term regional/rural is used broadly to refer to MMM categories two to seven.

The number of ophthalmologists working in each MMM category in each year was investigated and MMM category retention was determined based on MMM category during the six-year study period. The demographic characteristics of those who remained in MMM1 and those who remained in MMM2–MMM7 were investigated.

The location stability of the cohort of ophthalmologists who had a known postcode in 2014 was investigated using Kaplan-Meier survival analysis. A ‘failure’ event was defined as leaving MMM category (for any reason) and a censored event was defined as remaining in the same MMM category. The probability of ophthalmologists remaining in the same MMM category continuously for six years was calculated for those practicing in MMM1 and MMM2–MMM7.

As AHPRA did not provide registration status for 2019, a secondary competing-risks regression was conducted for the period 2014–2018, where registration status (e.g., current, did not renew, surrendered, deceased) was provided for each year. Registration status provides information on the reason for leaving MMM (e.g., either moving to another location or surrendering registration). Semiparametric competing-risks models, developed by Fine and Grey [20], were used to calculate the sub-hazards of leaving PPP MMM category to practice in another MMM or leaving for another reason (e.g., failed to renew, surrendered, withdrawn, or deceased, combined into one category). Sex, country of training (Australia/New Zealand vs. other) and years since primary qualification, were covariates in the preliminary models but were removed from the final models as they did not have significant effects on the incidence of moving MMM or leaving MMM for another reason. Stacked cumulative incidence plots (of moving MMM or leaving for another reason) were generated for ophthalmologists in MMM1 compared with MMM2–MMM7.

## 3. Results

The AHPRA dataset contained records for 1056 ophthalmologists. After excluding 108 records that did not specify at least one current registration and Australian location between 2014–2019, 948 ophthalmologists were included in the study (Table 1).

The state distribution of ophthalmologists remained stable over the study period, although there was a slight decrease in proportion of ophthalmologists located in South Australia (Table 2). Analysis by MMM category found the majority of ophthalmologists worked in MMM1 each year. However, there was a smaller proportion of ophthalmologists working in MMM1 in 2017–2019 compared with 2014–2016. While the numbers are small, there was a slight increase in the number of ophthalmologists working in MMM2 in 2019 compared with previous years.

There were 655 (69.1%) ophthalmologists who practiced continuously in MMM1 from 2014 to 2019 and 105 (11.1%) who practiced continuously in MMM2–MMM7 (Table 3). Among those who remained in MMM2–MMM7, 90.5% were male and 49% obtained their primary qualification 31–40 years ago. A total of 12 (1.3%) ophthalmologists moved from MMM1 to MMM2–MMM7. All 12 of these ophthalmologists were Australian graduates.

The Kaplan-Meier survival analysis for remaining in the same MMM from 2014–2019 included 896 ophthalmologists, with 4235 person-years at risk for ophthalmologists moving MMM category. Figure 1 presents the Kaplan-Meier survival estimates by MMM1 and MMM2–MMM7. The overall probability of ophthalmologists remaining in the same MMM category continuously from 2014 to 2019 was 0.84 (95% CI 0.82, 0.87), with a probability of 0.85 (95% CI 0.83, 0.88) for those practicing in MMM1 in the baseline year of 2014 and 0.79 (95% CI 0.71, 0.85) for those practicing in MMM2–MMM7.

Seven ophthalmologists were notified to be deceased up to 2018, the last year that AHPRA provided registration status. There were 20 ophthalmologists who relocated their primary place of practice to a different MMM category and 82 ophthalmologists who had a registration status of failed to renew, surrendered or deceased from 2014 to 2018. The competing-risks regression found the sub-hazard for moving MMM among ophthalmologists with a primary place of practice in MMM1 in 2014 was 24% (SHR 0.24, 95% CI 0.10, 0.59, *p* = 0.002) that of ophthalmologists with a primary place of practice in MMM2–MMM7. However, there was no significant difference in leaving MMM due to failure to renew, surrendered or deceased (SHR 1.08, 95% CI 0.58, 2.03, *p* = 0.80). Figure 2 presents the cumulative incidence plots of moving MMM or leaving MMM for another reason (e.g., failed to renew, surrendered, deceased) between 2014 and 2018, for ophthalmologists in MMM1 compared with MMM2–MMM7.

## 4. Discussion

The results indicate that 84% of ophthalmologists remained working in the same Modified Monash Model category from 2014 to 2019. Largely, these ophthalmologists were working in MMM1 compared to MMM2–MMM7 areas (85% vs. 75%).

The competing-risks regression models for the period 2014–2018 (the time period that included registration status data) showed that the hazard of moving MMM was higher among those in regional/rural areas than metropolitan areas. However, there was no difference between regional/rural and metropolitan ophthalmologists for the risk of leaving MMM due to not renewing registration due to temporary overseas relocation or surrendering registration due to retirement.

There was a trend for an increasing proportion of ophthalmologists to work outside major cities, from 19% in 2014 to 24% in 2019. This is similar to the finding of a United States study [21], where there was a mean annual increase of 2.3% in the density of ophthalmologists working in rural areas from 1995 to 2017.

Despite the trend for an increasing proportion of ophthalmologists working outside major cities, the workforce remains maldistributed. Unpublished Australian Bureau of Statistics population data show 72% of the Australian population lives in metropolitan (MMM1) areas [22]. This suggests that the concentration of ophthalmologists in major cities does not match the population distribution. This finding contrasts with a recent study conducted in New Zealand which found the number of ophthalmologists in each region was in proportion to the population size [23].

The majority of ophthalmologists working outside MMM1 areas are based in large regional centres (MMM2), not MMM3–MMM7 areas where the current level of ophthalmology services available seems insufficient. The disparity in access to ophthalmologists is evident in the finding by Keefe et al. that 15% of the population living in urban areas had ever seen an ophthalmologist compared with 2% of rural Australians [24]. This is despite higher prevalences of eye diseases and vision loss [1,2,3,4,5,6,7,8]. Policies that support strategies to increase the number of ophthalmologists in MMM3–MMM7 areas are needed to address the considerable disparity in access to ophthalmologists.

The cross-sectional analysis found a larger proportion of male ophthalmologists, compared with female ophthalmologists, stayed in MMM2–MMM7 areas. This finding is similar to that of Lo et al. who reported a lower proportion of female ophthalmologists compared with males working in regional/rural public facilities (18% vs. 27%) or private practices (18% vs. 40%) [25]. This may stem from female ophthalmologists being more likely to have a partner who is employed [25], which limits flexibility of work location. Indeed, opportunities for partner employment is a key factor in recruiting and retaining specialists in regional/rural areas [26,27].

Other studies have reported that a lack of suitable educational opportunities for children are a factor in specialist attrition from remote areas [26,27]. The finding that ophthalmologists who received their primary medical qualification 31–40 years ago were more likely to stay MMM2–MMM6 may reflect this age group being less likely to have young or school-aged children and therefore more content to remain regional/rural. This finding suggests that policies to attract new ophthalmologists to regional/rural areas may not be successful in the longer term, unless policies are implemented to help improve access to educational opportunities and the social integration of new ophthalmologists with young families to rural communities.

It is interesting that in the cross-sectional analysis, non-Australian/New Zealand medical graduates were over-represented in the group that remained in MMM2–MMM7 areas. The larger proportion of international graduates remaining in MMM2–MMM7 could be a policy outcome from the District of Workforce Shortage restrictions for international medical practitioners rather than a choice of the individual. Alternatively, it may be that a greater focus on regional/rural training for ophthalmology registrars is helping increase the attractiveness of practicing in regional/rural areas by demonstrating the more diverse and broader scope of practice available [28]. With the widespread availability of high-speed internet throughout Australia, ophthalmologists working in regional/rural areas are not so isolated anymore and have access to similar learning and networking opportunities as their metropolitan colleagues.

Study limitations include the combined analysis of MMM2–MMM7 categories. Whilst individual analysis of MMM categories may have allowed for greater insights regarding the rural and remote ophthalmology workforce, the small numbers of ophthalmologists working in discrete MMM3 or higher categories prevented further investigation due to the potential for identification of individuals. Fortunately, the Specialist Training Program (STP), funded by the Australian Commonwealth Government Department of Health, currently funds 15 training posts for ophthalmology in a range of private providers and MMM2 and above areas [28], therefore the current data provides key insight for this purpose.

Other limitations include the absence of data on outreach specialist ophthalmological services provided to remote communities and secondary places of practice. Unfortunately, this workforce information is not recorded by AHPRA. The provision of outreach services means that it is likely that the number of ophthalmologists working (even if on an intermittent basis) in MMM6–MMM7 categories is higher than the number identified in the AHPRA data. The lack of data on the number of years since specialists received their RANZCO fellowship was another limitation. Unfortunately, this was not available, only years since primary qualification. Lastly, the analysis was based on continuity of Modified Monash Model category. Ophthalmologists may have moved within or across states but remained in the same MMM category. These relocations were not considered important as the study was primarily focused on continuity of remoteness category, rather than physical location. Analysis of continuity of suburb location within MMM categories was not possible as the numbers became too small within sub-groups.

## 5. Conclusions

While the Australian ophthalmology workforce is skewed toward metropolitan centres, there are early indications of outward migration from metropolitan centres to regional/rural areas by Australian trained ophthalmologists. This reflects trends observed in the United States and New Zealand. Further longitudinal tracking of the primary work location of ophthalmologists is required to confirm the trend recognised in this study. Addressing workforce maldistribution problems will provide opportunities for regional/rural communities to better access ophthalmic services.

## Figures and Tables

**Figure 1 ijerph-18-12574-f001:**
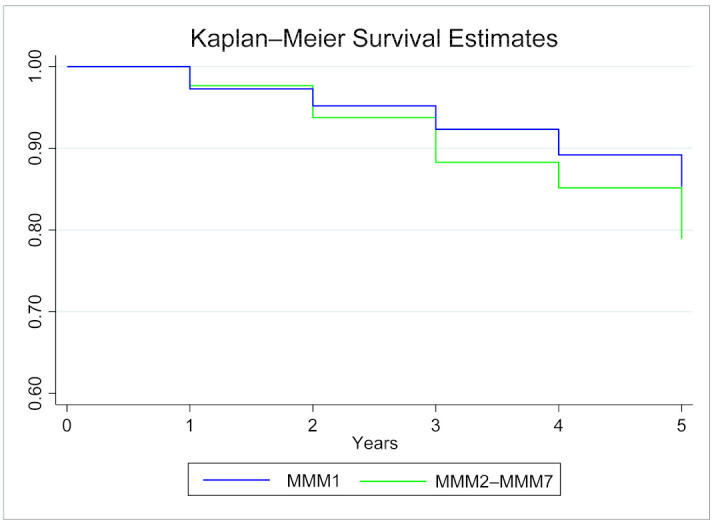
Kaplan-Meier Survival Estimates for staying practicing in MMM1 and MMM2–MMM7, 2014 to 2019.

**Figure 2 ijerph-18-12574-f002:**
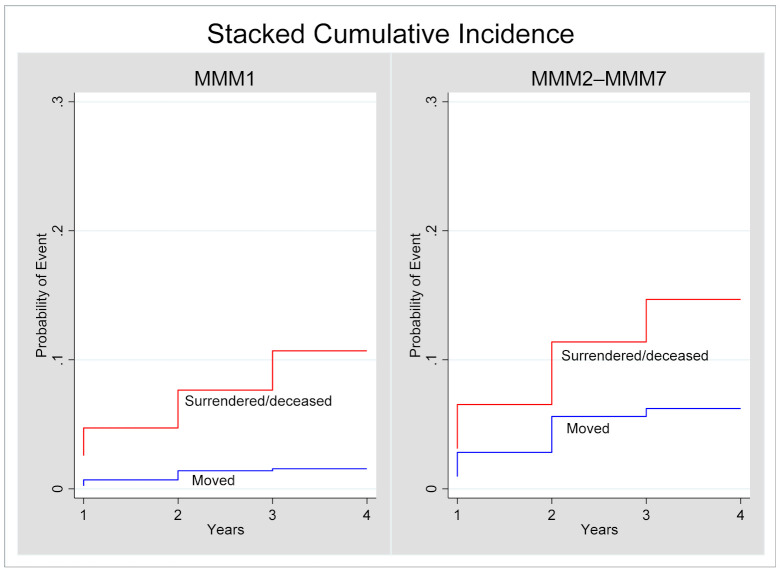
Stacked cumulative incidence plots for moving primary place of practice MMM or leaving due to other reason.

**Table 1 ijerph-18-12574-t001:** Demographic characteristics.

Characteristic	*n* = 948 (%)
Sex	
Male	767 (80.9)
Female	181 (19.1)
Years since primary qualification (median, IQR)	35 (25, 44)
Country of primary qualification	
Australia	746 (78.7)
United Kingdom	62 (6.5)
India	36 (3.8)
South Africa	32 (3.4)
New Zealand	29 (3.1)
Other	22 (2.3)
Unknown	21 (2.2)

**Table 2 ijerph-18-12574-t002:** Australian state or territory and MMM category by year (*n* = 948).

	2014	2015	2016	2017	2018	2019
**State**						
NSW	340 (35.9)	335 (35.3)	333 (35.1)	325 (34.3)	314 (33.1)	325 (34.3)
VIC	219 (23.1)	219 (23.1)	218 (23.0)	217 (22.9)	212 (22.4)	213 (22.5)
QLD	163 (17.2)	156 (16.5)	154 (16.2)	153 (16.1)	144 (15.2)	152 (16.0)
WA	71 (7.5)	73 (7.7)	68 (7.2)	66 (7.0)	66 (7.0)	70 (7.4)
SA	68 (7.2)	66 (7.0)	64 (6.8)	59 (6.2)	55 (5.8)	55 (5.8)
TAS	17 (1.8)	16 (1.7)	16 (1.7)	18 (1.9)	16 (1.7)	20 (2.1)
ACT	13 (1.4)	13 (1.4)	13 (1.4)	12 (1.3)	12 (1.3)	14 (1.5)
NT	5 (0.5)	5 (0.5)	4 (0.4)	4 (0.4)	4 (0.4)	6 (0.6)
**MMM Category**						
MMM1	768 (81.0)	757 (79.9)	744 (78.5)	723 (76.3)	698 (73.6)	721 (75.9)
MMM2	64 (6.8)	63 (6.6)	63 (6.6)	69 (7.3)	66 (7.0)	76 (8.0)
MMM3	55 (5.8)	54 (5.7)	54 (5.7)	52 (5.5)	51 (5.4)	52 (5.5)
MMM4	†	†	†	5 (0.5)	6 (0.6)	5 (0.5)
MMM5	†	†	†	†	†	†
MMM6	†	†	†	†	0	†
MMM7	0	0	0	0	0	0
Missing	52 (5.5)	65 (6.9)	78 (8.2)	94 (9.9)	125 (13.2)	93 (9.8)

Note: missing MMM data due to noncurrent registration for the year. † Omitted from cell as fewer than 5 persons. Percentages reported by column. NSW: New South Wales, VIC: Victoria, QLD: Queensland, WA: Western Australia, SA: South Australia, TAS: Tasmania, ACT: Australian Capital Territory, NT: Northern Territory.

**Table 3 ijerph-18-12574-t003:** Characteristics of ophthalmologists by geographic mobility group, 2014 to 2019.

	Stayed MMM1 (*n* = 655)	Stayed MMM2–MMM7 (*n* = 105)
Sex		
Male (*n* = 767)	514 (78.5)	95 (90.5)
Female (*n* = 181)	141 (21.5)	10 (9.5)
Years since primary qualification		
11–20 years (*n* = 87)	64 (9.9)	†
21–30 years (*n* = 276)	215 (33.2)	30 (28.8)
31–40 years (*n* = 260)	189 (29.2)	51 (49.0)
41–50 years (*n* = 192)	126 (19.5)	17 (16.3)
>50 years (*n* = 122)	53 (8.2)	†
Country of primary qualification		
Australia, NZ (*n* = 775)	549 (85.8)	81 (79.4)
Other (*n* = 152)	91 (14.2)	21 (20.6)

† Omitted from cell as fewer than 5 persons. Percentages reported by column.

## Data Availability

For access to the data please contact the corresponding author.

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
