# Peer review of "Distribution and Location Stability of the Australian Ophthalmology Workforce: 2014–2019"

_ijerph, 2021, doi:10.3390/ijerph182312574_

Round 1
Reviewer 1 Report
Reviewer Comments:
Thank you for the opportunity to review the manuscript "Distribution and Location Stability of the Australian Ophthalmology Workforce: 2014-2019".
This is an above average manuscript and worthy of publication. The introduction provides the reader with sufficient data to situate the reader in the subject matter. The method is described in a precise and concise manner.
When it comes to the results chapter, there are data that could be improved for its comprehension. In Table 2, in the first part of the table, the authors use a series of abbreviations that should be clarified. It would also be advisable to include a glossary with all the abbreviations.
Finally, the discussion and conclusions are appropriate to the results obtained.
Author Response
Thank you for taking the time to read our paper, you for your kind comments and helpful suggestions. We have spelled out the abbreviations below table 2. We have also added a glossary at the end of the reference section.
Reviewer 2 Report
Distribution and Location Stability of the Australian Ophthalmology Workforce: 2014-2019
The presented article is an excellent scientific work. The conclusions of this work are also of significant importance for society. Addressing workforce misdistribution problems will provide opportunities for regional/rural communities to better access ophthalmic services.
The conclusions of the study may be helpful for:
- health ministries to create incentive programs for ophthalmologists to mow rural locations,
- universities to sensitize ophthalmology students to the need to support rural areas,
- institutions increasing the professional competencies of ophthalmologists in terms of even greater use of online training opportunities (no barrier related to living outside metropolises).
In a formal matter: the method is selected correctly and correctly applied. In addition, it should be appreciated that the data to be analyzed is up-to-date, which significantly adds value to the work.
It is interesting to compare the results obtained with the results for Australia to the results from the United States of America and New Zealand.
I have no objections. Congratulations to the Authors of a good article.
Author Response
Thank you for taking the time to read our paper and thank you for your kind comments.
Reviewer 3 Report
The authors of this manuscript describe the distribution and location stability of the Australian Ophthalmology Workforce from 2014 until 2019. The introduction summarizes important literature and methods are explained well. The manuscript is well written and addresses an important topic as there is a need for policy initiatives to train, recruit, and retain eye specialists in regional, rural and remote areas.
Minor suggestions are as follows;
In my opinion, a few claims in the paper might benefit from reference citations e.g. the very first sentence.
Methods section; perhaps the authors might consider including the estimated resident population when describing MMM1-MMM7 as it would provide good context for readers that live outside Australia.
Author Response
Thank you for taking the time to read and provide helpful comments to improve our paper.
In the introduction section we have moved citations forward to the first relevant sentence.
We agree that it would be helpful to add information about the population size within each MMM category. Unfortunately, the Australian Bureau of Statistics has not published this information. The only information regarding population size by MMM category is available in the Factsheet on the Australian Department of Health Modified Monash Model website. The Factsheet states the vague information that '70% of the population lives in MMM1'. We have added this information to the methods and included the Factsheet citation.